Detection of diabetic macular oedema: validation of optical coherence tomography using both foveal thickness and intraretinal fluid

Hernández-Martínez Carmen 1
Palazón-Bru Antonio 2 3 antonio.pb23@gmail.com
Azrak Cesar 1
Navarro-Navarro Aída 1
Baeza-Díaz Manuel Vicente 1 2
Martínez-Toldos José Juan 1
Gil-Guillén Vicente Francisco 2 3
1 Ophthalmology Service, General Hospital of Elche , Elche, Alicante , Spain
2 Department of Clinical Medicine, Miguel Hernández University , San Juan de Alicante, Alicante , Spain
3 Research Unit, Elda Hospital , Elda, Alicante , Spain
Fleiszig Suzanne
Electronic publication date: 2015 Nov 10
Publication date: 2015
Volume: 3
Electronic Location ID: e1394
Received 2015 Jun 25; Accepted 2015 Oct 19
Copyright: © 2015 Hernández-Martínez et al.
Copyright year: 2015
Copyright holder: Hernández-Martínez et al.
License: This is an open access article distributed under the terms of the Creative Commons Attribution License, which permits unrestricted use, distribution, reproduction and adaptation in any medium and for any purpose provided that it is properly attributed. For attribution, the original author(s), title, publication source (PeerJ) and either DOI or URL of the article must be cited.
License URL: https://creativecommons.org/licenses/by/4.0/

Keywords: Macular oedema, Optical coherence tomography, Diabetes complications, Vision screening

Funding: The authors received no funding for this work.

==============================
No studies have yet evaluated jointly central foveal thickness (CFT) and the presence of intraretinal fluid (PIF) to diagnose diabetic macular oedema (DMO) using optic coherence tomography (OCT). We performed a cross-sectional observational study to validate OCT for the diagnosis of DMO using both CFT and PIF assessed by OCT (3D OCT-1 Maestro). A sample of 277 eyes from primary care diabetic patients was assessed in a Spanish region in 2014. Outcome: DMO diagnosed by stereoscopic mydriatic fundoscopy. OCT was used to measure CFT and PIF. A binary logistic regression model was constructed to predict the outcome using CFT and PIF. The area under the ROC curve (AUC) of the model was calculated and non-linear equations used to determine which CFT values had a high probability of the outcome (positive test), distinguishing between the presence or absence of PIF. Calculations were made of the sensitivity, specificity, and the positive (PLR) and negative (NLR) likelihood ratios. The model was validated using bootstrapping methodology. A total of 37 eyes had DMO. AUC: 0.88. Positive test: CFT ≥90 µm plus PIF (≥310 µm if no PIF). Clinical parameters: sensitivity, 0.83; specificity, 0.89; PLR, 7.34; NLR, 0.19. The parameters in the validation were similar. In conclusion, combining PIF and CFT provided a tool to very precisely discriminate the presence of DMO. Similar studies are needed to provide greater scientific evidence for the use of PIF in the diagnosis of DMO.

Introduction

Diabetes mellitus affects around 180 million people worldwide (WHO, 2002). Two important complications of the disease result in loss of vision, proliferative diabetic retinopathy and diabetic macular oedema. The latter is the leading cause of reduced vision in diabetic patients (The Diabetic Retinopathy Study Research Group, 1987; Aiello, 2003).

The diagnostic method of reference (Clinical Standard) for diabetic macular oedema is based on the stereoscopic examination of the fundus of the eye using a magnifying lens (with or without making contact) to visualise the retina with mydriasis (Early Treatment Diabetic Retinopathy Study Research Group, 1985; Kinyoun et al., 1989). Other methods, more common in primary care, are direct ophthalmoscopy or non-stereoscopic retinography. However, diffuse thickening or thickening in the form of retinal cysts as the initial sign of diabetic macular oedema may go unnoticed with these examination methods, as they can only detect macular oedema from the presence of hard exudates or indirect signs, such as macular haemorrhage or microaneurysms (Aldington et al., 1995; Bursell et al., 2001; Gómez-Ulla et al., 2002; Rudinsky et al., 2002; Herbert, Jordan & Flanagan, 2003; Baeza Diaz et al., 2004; Baeza et al., 2009).

The introduction of optical coherence tomography (OCT) led to changes in both the diagnosis and the management of diabetic macular oedema. OCT can provide high-resolution cross-sectional images of the retina. These can give us information about the retinal thickness and its morphology, as well as whether or not there is intraretinal fluid, key information in the diagnosis of diabetic macular oedema (Hee et al., 1995; Hee et al., 1998; Koozekanani et al., 2000; Browning et al., 2004; Virgili et al., 2015).

Some OCT devices can provide an image of the retina as well as a cross-sectional view. These two observations constitute an excellent tool for the diagnosis of macular oedema. Accordingly, the use of OCT can be considered as a screening method for diabetic macular oedema in primary care, given that it solves the problems associated with the previously described methods. However, as studies assessing the validity of OCT as a diagnostic method for macular oedema are lacking, we undertook a study to validate OCT (3D OCT-1 Maestro; Topcon Corporation®, Itabashi, Tokyo, Japan) for the diagnosis of diabetic macular oedema.

Materials & Methods

Study population

Diabetic patients who attended their primary healthcare centre in Elche (Spain), an industrial city with a medium socioeconomic status.

Study design and participants

This cross-sectional observational study involved a sample of diabetic patients (types 1 and 2) who attended their primary healthcare centres (in the General Hospital of Elche catchment area) between January and May 2014 for screening for diabetic retinopathy and diabetic macular oedema. Exclusion criteria were at least one of the following: another macular disorder, high myopia, dementia, a cataract operation during the previous three months, vitreoretinal surgery, laser treatment in the macular area or panphotocoagulation, and anti-angiogenic drugs.

Variables and measurements

The main study outcome variable was the presence of diabetic macular oedema, determined by stereoscopic mydriatic fundoscopy using a slit lamp with a non-contact magnifying lens (78D aspheric lens; Volk Optical Incorporated Company, Mentor, OH), performed by an expert retinal ophthalmologist (Early Treatment Diabetic Retinopathy Study Research Group, 1985; Kinyoun et al., 1989).

Using the OCT that we wished to validate, we obtained information about the central foveal thickness (µm) and the presence of intraretinal fluid or cysts (hyporeflective areas). The central foveal thickness was obtained after dilation of the pupil with a drop of tropicamide. The images were acquired as a 3D 6 mm × 6 mm volume cube. The mean retinal thickness was calculated automatically by the device software. This was used to evaluate the central circle (1,000 µm in diameter). In addition, we used the OCT to obtain a horizontal tomographic image of the retina (B-Scan), in which we assessed the presence or otherwise of intraretinal fluid (cysts or hyporeflective intraretinal areas) (Fig. 1).

Figure 1 Screenshot of the retinal map analysis.

The presence of intraretinal fluid is seen in the upper left image (B-scan). Both the fundus image and the central foveal thickness are shown in the upper right image. This figure is published with permission from Topcon Corporation®.

Finally, and for descriptive purposes only, records were made of gender, type of diabetes mellitus, insulin use, vascular disease, stroke, coronary heart disease, hypertension, dyslipidaemia, smoking, age (years), years since diagnosis of diabetes, body mass index (BMI) (kg/m2), HbA1c (%) and best corrected visual acuity. These variables were determined from a clinical interview and corroborated by the charts, except for BMI, HbA1c and visual acuity, which were measured using standard methods (American Diabetes Association, 2014).

Sample size

The final sample comprised 277 eyes, of which 37 showed diabetic macular oedema. In order to contrast an area under the ROC curve (AUC) different from 0.5 (the tested variable does not have the discriminatory power to identify diabetic macular oedema in the patients), assuming 95% confidence and an expected AUC of 0.85, the statistical power obtained was almost 100% (Hanley & McNeil, 1982).

Statistical methods

The variables are described as absolute and relative frequencies (qualitative variables) or means and standard deviations (quantitative variables). A binary logistic regression model using generalized estimating equations (to take into account that we could have two values in the data set for each patient) was constructed with macular oedema (Clinical Standard) as the dependent variable, and the central foveal thickness and the presence of intraretinal fluid as the independent variables. The goodness-of-fit of this model was assessed with the likelihood ratio test. The prognostic probabilities of this multivariate model and their AUC were calculated. The optimal point for the probabilities was determined (that which minimized the square root of (1-sensitivity)2 +(1-specificity)2) and non-linear equations were used to determine which values of foveal thickness presented a prognostic probability above the optimal point, distinguishing between the presence or absence of intraretinal fluid. Finally, calculations were made of the sensitivity, specificity, and the positive (PLR) and negative (NLR) likelihood ratios. To validate the model, we performed 1,000 bootstrapping samples to calculate: sensitivity, specificity, PLR, NLR and AUC (El Maaroufi et al., 2015). The analyses were all done with a significance of 5% and the confidence interval (CI) was calculated for each relevant parameter. The calculations were done with IBM SPSS Statistics 19 and R 2.13.2.

Ethical considerations

The study was approved by the Ethics Committee of Elche Hospital and all the participants gave written informed consent (final approval date: February 24, 2014). All the personal information was deleted during the statistical analysis process. The study complies with the Declaration of Helsinki.

Results

Of the 140 patients (280 eyes) seen during the study period, 3 eyes were excluded (1, vitreomacular traction syndrome; 1, macular degeneration; 1, epiretinal membrane). Of the remaining 277 eyes, 37 had macular oedema (13.36%, 95% CI [9.35–17.36]). The mean central foveal thickness was 270.4 µm and 17.3% of the patients had intraretinal fluid according to the OCT. These values and the other descriptive characteristics can be seen in Table 1.

Table 1 Descriptive characteristics of the eyes of the diabetic patients studied.

Variable	Total (n = 277) n(%)/x ± s	
Male gender	162(58.5)	
Type 2 diabetes mellitus	249(89.9)	
Insulin use	125(45.1)	
Vascular disease	46(16.6)	
Stroke	6(2.2)	
Coronary heart disease	42(15.2)	
Hypertension	137(49.5)	
Dyslipidaemia	141(50.9)	
Smoking	49(17.7)	
Intraretinal liquid	48(17.3)	
Foveal thickness (µm)	270.4 ± 45.1	
Age (years)	61.6 ± 13.0	
Years since diabetes diagnosis	14.1 ± 10.8	
BMI (kg/m2)	28.8 ± 4.9	
HbA1c (%)	7.5 ± 1.7	
Visual acuity	0.85 ± 0.20	
Notes.

BMI body mass index

n(%) absolute frequency (relative frequency)

x ± s mean ± standard deviation

The logistic regression model was highly significant (X2 = 141.5, p < 0.001) and gave an AUC of 0.88 (95% CI [0.82–0.95], p < 0.001) (Fig. 2). The equation for the prognostic probabilities can be found in Supplemental Information 1. The cut points of central foveal thickness according to the presence or absence of intraretinal fluid are shown in Fig. 3. This figure shows that if the patient has intraretinal fluid, the test will be positive if the foveal thickness is ≥90 µm, whereas if there is no intraretinal fluid the cut point is 310 µm. This diagnostic test had the following clinical parameters: sensitivity, 0.83 (95% CI [0.69–0.92]); specificity, 0.89 (95% CI [0.84–0.92]); PLR, 7.34 (95% CI [5.00–10.78]); and NLR, 0.19 (95% CI: 0.10–0.36). The validation parameters using the bootstrapping methodology were: sensitivity, 0.83 (95% CI [0.71–0.93]), specificity, 0.89 (95% CI: [0.85–0.93]); PLR, 7.28 (95% CI [5.18–11.19]); NLR, 0.19 (95% CI [0.09–0.33]); AUC, 0.89 (95% CI [0.80–0.94]).

Figure 2 ROC curve of the prognostic probabilities of the multivariate model constructed.

AUC, area under the ROC curve; CI, confidence interval.

Figure 3 Cut points obtained for the diagnosis of diabetic macular oedema according to the presence or absence of intraretinal fluid.

Discussion

Summary

This study determined cut points for foveal thickness taking into account the presence of intraretinal fluid. In addition, the joint evaluation of these two parameters gave a high discriminating power to differentiate between patients who have macular oedema and those who do not.

Strengths and limitations of this study

A literature search failed to detect any studies that evaluated jointly the two parameters analysed here (foveal thickness and intraretinal fluid) to detect diabetic macular oedema. Accordingly, our results are novel. The statistical power with the sample used was nearly 100% and the discrimination of the mathematical model was very high (AUC > 0.85). Finally, the OCT validated in this study is very simple to use, as it automatically focuses on the retina and does all the calculations; the user only has to look at them and make a clinical decision.

We attempted to minimise selection bias by using a sample obtained via a random design. Concerning information bias, all the variables were collected by expert ophthalmologists.

Comparison with the existing literature

In 2015, Virgili et al. published a meta-analysis analysing the discriminatory power of foveal thickness to diagnose diabetic macular oedema. Calculation of the summary parameters in this meta-analysis was: sensitivity, 0.81; specificity, 0.85; PLR, 5.4; NLR, 0.22. With these parameters, we can calculate the summary distance on the ROC curve from the optimal cut point of foveal thickness to the left top vertex (sensitivity = 1 and specificity = 1), obtaining a value of 0.2421. Our study, though, gave a distance of just 0.2025. In other words, the discriminating power of our model is greater than that of the other studies; therefore, adding the presence of intraretinal fluid improves the ability to detect diabetic macular oedema.

Implications for clinical practice and research

Analysis of our results and their comparison with other studies shows that the presence of intraretinal fluid should definitely be taken into account when diagnosing macular oedema. Moreover, the diagnostic test designed here does not just determine one single cut point, but rather it constructs intervals of foveal thickness depending on the presence of intraretinal fluid.

Given that we have been unable to find any other studies that analysed jointly foveal thickness and the presence of intraretinal fluid, we encourage others to determine whether these results hold true in other populations with a different prevalence of macular oedema. If this is found to be the case, we shall then have a more precise screening tool for macular oedema than currently used (just foveal thickness).

Conclusion

Combining the presence of intraretinal fluid with central foveal thickness provides a tool to discriminate precisely between diabetic patients with and without macular oedema. Furthermore, given that the OCT used in this study is very simple to use and to interpret, it could become a screening tool in primary healthcare. Finally, the very satisfactory results of this study suggest that similar studies should also be done to provide greater scientific evidence for the use of intraretinal fluid in the diagnosis of macular oedema.

Supplemental Information

Supplemental Information 1 Predictive model

FT, foveal thickness; IRF, intraretinal fluid.

Click here for additional data file.

Supplemental Information 2 Raw data

Click here for additional data file.

The authors thank Maria Repice and Ian Johnstone for help with the English language version of the text.

Additional Information and Declarations

Competing Interests

Author Contributions

Human Ethics

Antonio Palazón-Bru is an Academic Editor for PeerJ.

Carmen Hernández-Martínez conceived and designed the experiments, performed the experiments, wrote the paper.

Antonio Palazón-Bru conceived and designed the experiments, analyzed the data, wrote the paper, prepared figures and/or tables.

Cesar Azrak conceived and designed the experiments, wrote the paper.

Aída Navarro-Navarro and Manuel Vicente Baeza-Díaz conceived and designed the experiments, performed the experiments, reviewed drafts of the paper.

José Juan Martínez-Toldos conceived and designed the experiments, contributed reagents/materials/analysis tools, reviewed drafts of the paper.

Vicente Francisco Gil-Guillén conceived and designed the experiments, reviewed drafts of the paper.

The following information was supplied relating to ethical approvals (i.e., approving body and any reference numbers):

The study was approved by the Ethics Committee of Elche Hospital and all the participants gave written informed consent (final approval date: February 24, 2014). All the personal information was deleted during the statistical analysis process. The study complies with the Declaration of Helsinki.

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
