# Peer review of "Detection of diabetic macular oedema: validation of optical coherence tomography using both foveal thickness and intraretinal fluid"

_PeerJ, doi:10.7717/peerj.1394_

## Round 0.1 · original submission · Minor Revisions

· Academic Editor

Minor Revisions

The reviewers both believe that this is an informative study. Both have suggested minor revisions that include clarification of what was done and how the data were analyzed.

Reviewer 1 ·

Basic reporting

This paper is well written and the study appears to have been conducted well. The findings are impressive and straightforward and the objective of the paper is clear.

Experimental design

In general, the study appears to have been well designed and conducted carefully. With the addition of a few revisions, this can be an informative paper.

Validity of the findings

Please see general comments. The validity of this approach and this model would be improved by performing some type of assessment of these data (eg developing the model on half of a randomly selected group of patients, and testing the model on the second half, or alternatively using cross-validation, resampling, bootstrapping or some other means of determining how robust the model appears to be).

Additional comments

I believe that this is a paper that would be of value and interest to readers of this journal. I have just a few comments and suggestions for this paper.

I would recommend replacing "gold standard" with "clinical standard". A gold standard implies an absolute objective measure, and this comparison involves a subjective determination by expert clinicians.

Although the validity and generalizability of this work will ultimately be determined by other cventers using the model and assessing performance, it would also be helpful to use various techniques for evaluating the model with the existing data. One could use cross-validation, bootstrapping or resampling to evaluate the model, or develop the model on one half (randomly selected) of the patients, and testing the model on the other half.

Reviewer 2 ·

Basic reporting

No Comments

Experimental design

Additional information needs to be provided regarding the statistical modelling. Specifically, whether the nesting of eye within subject was taken in to account while constructing the statistical models.

Validity of the findings

No Comments.

Additional comments

Specific comments related either to wording, clarity of exposition or statistical methods.

Introduction

Lines 18-20 don’t make much sense. The phrase ‘stereoscopic examination of the fundus of the eye using a magnifying lens.. to visualise the pupil in mydriasis.’ doesn’t seem right. If you are examining the retina then how can you be visualizing the pupil?

Line 28. “optic coherence tomography” should be “optical coherence tomography”.

Line 34. OCT doesn’t really provide a ‘photograph’ of the retina. It is perhaps more correct to call it an ‘en face image’ or just an ‘image’ of the retina.


Methods

Line 61. I’m assuming 6x6 should be 6 degrees by 6 degrees. Is that correct?

Lines 62 and 63. Is the ‘central 1000 micron area’ actually a central circle with diameter = 1000 microns?

Sample size section. How did you come up with and expected AUC of 0.95 when estimating your power for detecting a difference from 0.5?

Statistical methods section. Did the binary logistic regression model take account of the clustering of eyes within individual? The model generated can be in error if a known level of nesting / clustering is ignored, especially the p-value (given later in line 103). Consider using Generalized Estimating Equations (GEE) with binary logistic dependent variable, which can be performed in SPSS Statistics 19.

Line 85. Instead of ‘points of foveal thickness’ consider ‘values of foveal thickness’.


Discussion
Line 124. Take a decision or make a decision?


Figure captions. Figure 1, consider changing ‘fundus photograph’ to ‘fundus image’.

---

## Round 0.2 · accepted · Accept

· Academic Editor

Accept

The reviewers are satisfied with the revised version of this manuscript.

Reviewer 1 ·

Basic reporting

The authors have appropriately responded to the comments and suggestions in the previous review, so I feel that this is now suitable for the readership of peerJ.

Experimental design

See above comments

Validity of the findings

See above comments

Additional comments

Revision looks fine

Reviewer 2 ·

Basic reporting

No Comments

Experimental design

No Comments

Validity of the findings

No Comments

Additional comments

No further suggestions